# Downregulation of miR-1388 Regulates the Expression of Antiviral Genes via Tumor Necrosis Factor Receptor (*TNFR*)-Associated Factor 3 Targeting Following poly(I:C) Stimulation in Silver Carp (*Hypophthalmichthys molitrix*)

**DOI:** 10.3390/biom14060694

**Published:** 2024-06-14

**Authors:** Kun Gao, Meng Liu, Huan Tang, Zhenhua Ma, Hanyu Pan, Xiqing Zhang, Muhammad Inam, Xiaofeng Shan, Yunhang Gao, Guiqin Wang

**Affiliations:** College of Animal Science and Technology, Jilin Agricultural University, Changchun 130118, China; kungao213@163.com (K.G.); liumeng4610@163.com (M.L.); tanghuan202308@163.com (H.T.); mazhenhua1030@163.com (Z.M.); panhanyu0407@163.com (H.P.); zhangxiqing1020@163.com (X.Z.); dr.inam@sbbu.edu.pk (M.I.); sxf1997@163.com (X.S.)

**Keywords:** silver carp, miR-1388, poly(I:C), *TRAF3*, innate immunity

## Abstract

MicroRNAs (miRNAs) are highly conserved endogenous single-stranded non-coding RNA molecules that play a crucial role in regulating gene expression to maintain normal physiological functions in fish. Nevertheless, the specific physiological role of miRNAs in lower vertebrates, particularly in comparison to mammals, remains elusive. Additionally, the mechanisms underlying the control of antiviral responses triggered by viral stimulation in fish are still not fully understood. In this study, we investigated the regulatory impact of miR-1388 on the signaling pathway mediated by IFN regulatory factor 3 (*IRF3*). Our findings revealed that following stimulation with the viral analog poly(I:C), the expression of miR-1388 was significantly upregulated in primary immune tissues and macrophages. Through a dual luciferase reporter assay, we corroborated a direct targeting relationship between miR-1388 and tumor necrosis factor receptor (*TNFR*)-associated factor 3 (*TRAF3*). Furthermore, our study demonstrated a distinct negative post-transcriptional correlation between miR-1388 and *TRAF3*. We observed a significant negative post-transcriptional regulatory association between miR-1388 and the levels of antiviral genes following poly(I:C) stimulation. Utilizing reporter plasmids, we elucidated the role of miR-1388 in the antiviral signaling pathway activated by *TRAF3*. By intervening with siRNA-*TRAF3*, we validated that miR-1388 regulates the expression of antiviral genes and the production of type I interferons (*IFN-Is*) through its interaction with *TRAF3*. Collectively, our experiments highlight the regulatory influence of miR-1388 on the *IRF3*-mediated signaling pathway by targeting *TRAF3* post poly(I:C) stimulation. These findings provide compelling evidence for enhancing our understanding of the mechanisms through which fish miRNAs participate in immune responses.

## 1. Introduction

Innate immunity, as the host’s first line of defense against pathogen invasion, is capable of rapidly recognizing and eliminating non-self-entities. Pattern recognition receptors (PRRs) recognize viral RNA, triggering the secretion of *IFN-I*, pro-inflammatory cytokines, and other responses. Toll-like receptors (*TLRs*) and retinoic acid-inducible gene I (*RIG-I*)-like receptors (*RLRs*) are major PRRs involved in viral surveillance. The activation of *TLRs* or *RLRs* leads to the activation of the transcription factor *IRF3*, regulating *IFN-I* secretion [1,2]. *IFN-I* is critical in antiviral defense across species [3,4,5], activating interferon-stimulated genes (*ISGs*), thereby maintaining the host’s immune state. As non-coding genes continue to be explored, their role in immune regulation through target gene regulation has garnered increasing attention.

Studies have shown widespread *TRAF3* expression in various tissues [6], indicating its broad physiological role. Reports indicate that *TRAF3* plays a crucial role in conditions, such as herpes simplex encephalitis, influenza A virus infection, and primary immunodeficiency [7,8,9]. As a critical immune gene, *TRAF3* activation ultimately triggers the essential transcription factor *IRF3*, augmenting *IFN-I* expression [7]. While *IRF3* is expressed at basal levels in normal cells, it undergoes rapid phosphorylation and activation upon viral invasion, contributing to the regulation of interferon and the expression of *ISGs* [10]. Moreover, *TRAF3* is a distinct member among *TRAF* family molecules and prominently functions in *TLR* and *RLR*-mediated antiviral responses [11,12]. *TRAF3* can be recruited by TIR-domain-containing adapter-inducing interferon-β (*TRIF*) or the mitochondrial antiviral signaling protein (*MAVS*) following the effective recognition of viral RNA, subsequently activating interferon regulatory factors to stimulate *IFN-I* production and combat viral intrusion.

MicroRNAs (miRNAs) constitute a class of highly conserved, approximately 22 nucleotide-long, single-stranded, non-coding RNA molecules in eukaryotic organisms, which modulate the expression of messenger RNA (mRNA) at the post-transcriptional level through targeted interactions with 3′ untranslated regions (3′UTR) and their corresponding mRNAs. Research has shown that miRNAs play crucial roles in growth, immunity, and cell death regulation [13,14,15]. The silver carp (*Hypophthalmichthys molitrix*) is an economically important freshwater fish species widely cultured in China, which faces unresolved challenges from viral diseases affecting aquaculture environments. Exposure to exogenous harmful substances can induce the upregulation or downregulation of miRNA expression in fish. Previous studies have identified miRNAs, such as miR-210, miR-155, and miR-126, as participating in antiviral immune responses [16,17,18]. A single miRNA can target multiple mRNAs, and conversely, a single mRNA can be regulated by multiple miRNAs, highlighting the broad regulatory effects of miRNAs on mRNA expression. miR-1388 has been demonstrated to play important roles in regulating cell junctions in flounder (*Parallichthys olivaceus*) [19], bacterial infections in yellow catfish (*Pelteobagrus fulvidraco*) [20], erythropoiesis in Antarctic icefish *Antarcticice*) [21], and redox reactions in Japanese flounder [22]. Using a dual-luciferase reporter system and green fluorescent protein assay, we confirmed the targeting relationship between miR-1388 and *TRAF3* in silver carp. Furthermore, our research elucidated the regulatory role and mechanism of miR-1388 in silver carp following poly(I:C) stimulation. These findings enhance our understanding of the critical role of miRNAs in innate immunity and provide new insights into the innate immune function of miR-1388 in silver carp.

## 2. Materials and Methods

### 2.1. Silver Carp and Treatment

Clinically healthy silver carp weighing approximately 150 g were procured from a commercial fish farm located in Jilin province, China. Prior to the commencement of the experiment, the fish were acclimated in indoor water circulation systems for a minimum of 5 weeks at a temperature of approximately 26–28 °C. The fish were randomly divided into the following two groups: the stimulation group and the control group. The stimulation group was intraperitoneally injected with 300 μL of poly(I:C) (1 mg/mL; InvivoGen, San Diego, CA, USA), while the control group was given an equal volume of physiological saline. Following 24 h of stimulation, various tissues (kidney, liver, heart, spleen, brain, and gill) of the silver carp were collected under sterile conditions after euthanization and promptly frozen in liquid nitrogen. The experimental protocol was approved by the Ethics Committee of Jilin Agricultural University (JLAU08201409), and all animal experiments were conducted in strict accordance with the U.K. Animals (Scientific Procedures) Act, 1986, and associated guidelines, as well as EU Directive 2010/63/EU for animal experiments.

### 2.2. Cell Culture, Macrophage Isolation and poly(I:C) Exposure

To isolate silver carp head kidney macrophages, we initially aseptically harvested the head kidneys from silver carp. Subsequently, these tissues were minced into small fragments using sterile shears and processed through a 70 µm cell strainer. We added separation solutions 1 and 2 into the centrifuge tube, followed by the addition of the cell homogenate. We perform gradient centrifugation (500× *g* for 25 min) to separate the intermediate layer of cells. The collected cells were then washed with precooled L-15 medium Hyclone, Logan, UT, USA) and centrifuged at 250× *g* for 10 min at 4 °C. This washing step was repeated once more for washing. After discarding the supernatant, the cell pellet was resuspended in L-15 medium supplemented with 1% dual antibiotics (100 IU/mL penicillin (Beyotime, Shanghai, China), 100 mg/mL streptomycin (Beyotime, Shanghai, China) and 2% fetal bovine serum (FBS) (Gibco, San Diego, CA, USA). The isolated macrophages were then seeded in 6-well plates at a density of 3.6 × 10^7^ cells per well and incubated at 25 °C under 5% CO_2_. The next day, the medium was replaced with fresh L-15 containing 15% FBS. After stimulation with 1 mL poly(I:C) (1 mg/mL), the cellular RNA or protein was extracted at various time points. Each experiment was performed in triplicate. Epithelioma papulosum cyprini (EPC) cells and human embryonic kidney (HEK293T) cells were procured fromShanghai Cell Bank. EPC cells were cultured in M199 medium (Hyclone, Logan, UT, USA) supplemented with 10% FBS and maintained at 26 °C under 5% CO_2_, while HEK293T cells were cultured in DMEM (Hyclone, Logan, UT, USA) supplemented with 10% FBS and maintained at 37 °C under 5% CO_2_.

### 2.3. Plasmid Construction and Transfection

To construct the *TRAF3* expression plasmid, the *TRAF3* (CDS+3′UTR) sequence with homology to the vector was initially amplified using the ClonExpress II One Step Cloning Kit (Vazyme, Nanjing, China) and then recombined with a pre-linearized pcDNA3.1-Flag vector (HindIII, XhoI). Additionally, the 3′UTR of *TRAF3* was inserted into the pmir-GLO vector to construct the wild *TRAF3* 3′UTR reporter plasmid. The mutant *TRAF3* 3′UTR was generated by synthesizing specific primers to introduce mutations in the *TRAF3* 3′UTR using the Mut Express II Fast Mutagenesis Kit V2 (Vazyme, Nanjing, China). Similarly, wild-type and mutant mVenus-*TRAF3*-3′UTR plasmids were constructed using the same method. All plasmids were confirmed by Sanger sequencing and purified using the EndoFree Maxi Plasmid Kit (TIANGEN, Beijing, China). Plasmid transfection was conducted using Lipofectamine 3000^TM^ (Invitrogen, Carlsbad, CA, USA) transfection reagents following the manufacturer’s protocol. The primer sequences are provided in Appendix A.

### 2.4. miRNA Mimics and Inhibitors

The miR-1388 mimics, inhibitor, and their corresponding negative control sequences were commercially synthesized by Genepharma (Suzhou, China) and are described as follows: miR-1388 mimics, sense: 5′-AGGACUGUCCAACCUGAGAAUG-3′, antisense: 5′-UUCUCAGGUUGGACAGUCCUUU-3′; mimics negative control, sense: 5′-UUCUCCGAACGUGUCACGUTT-3′, antisense: 5′-ACGUGACACGUUCGGAGAATT-3′; miR-1388 inhibitor, sense: 5′-CAUUCUCAGGUUGGACAGUCCU-3′; inhibitor negative control, 5′-CAGUACUUUUGUGUAGUACAA-3′.

### 2.5. RNA Interference

Small interfering RNAs (siRNAs) targeting *TRAF3* and their corresponding controls were synthesized by Genepharma (Suzhou, China) using the following sequences: si-TRAF3, sense: 5′-GGUAGAAACACAGUAUGAATT-3′, antisense: 5′-UUCAUACUGUGUUUCUACCTT-3′; negative control, sense: 5′-UUCUCCGAACGUGUCACGUTT-3′, antisense: 5′-ACGUGACACGUUCGGAGAATT-3′.

### 2.6. RNA Extraction and Real-Time Quantitative PCR

Total RNA was extracted from cells using the Trizol Reagent (Invitrogen) following the manufacturer’s instructions. The extracted RNA was then reverse-transcribed into cDNA using the TransScript^®^ Uni All-in-One First-Strand cDNA Synthesis SuperMix for the qPCR (One-Step gDNA Removal) kit (TransGen Biotech, Beijing, China). The resulting cDNA was used for quantification of the target gene using the PerfectStart^®^ Green qPCR SuperMix (TransGen Biotech, Beijing, China). For miRNA analysis, the miRNA 1st Strand cDNA Synthesis Kit (by stem-loop) (Vazyme, Nanjing, China) was used for the reverse transcription of miRNAs. The quantification of miRNAs was performed using the miRNA Universal SYBR qPCR Master Mix as well (Vazyme, Nanjing, China). Both mRNA and miRNA quantification were carried out on a LightCycler^®^ 96 real-time PCR system (Roche, Basel, Switzerland). U6 and GAPDH were used as endogenous controls for miRNA and mRNA quantification, respectively. The 2^−ΔΔCt^ method was used for analysis. All the values are presented as the mean ± SE. Please refer to Appendix A for the specific primer sequences used in this experiment.

### 2.7. Dual-Luciferase Reporter Assays

HEK293T cells were co-transfected with *TRAF3* 3′UTR WT or *TRAF3* 3′UTR MT plasmid, together with either miR-1388 mimics, the mimics NC inhibitor or inhibitor NC using Lipofectamine3000TM (Invitrogen, Carlsbad, CA, USA). Additionally, HEK293T cells were co-transfected with *TRAF3* overexpression plasmid, the *IRF3* or *IFN* stimulated response element (*ISRE*) luciferase reporter plasmids, and PhRL-TK Renilla luciferase plasmid, along with either miR-1388 mimics and mimics NC. After 24 or 48 h of transfection, cells were lysed, and the dual luciferase reporter assay kit (Vazyme, Nanjing, China) was used to measure all luciferase activity. The results were normalized to the Renilla luciferase control, and all experiments were performed in triplicate.

### 2.8. Western Blotting

The bicinchoninic acid (BCA) assay (Beyotime, Beijing, China) was used to determine the protein concentration in the supernatant of cell extracts after centrifugation, and the 1× SDS-PAGE Protein loading buffer (Beyotime, Shanghai, China) was used to equalize protein concentrations. Equal amounts of the extracts were then subjected to SDS-PAGE and transferred onto polyvinylidene difluoride (PVDF) membranes (Millipore, Boston, Massachusetts, USA). Membranes were incubated at 4 °C overnight with an anti-Flag rabbit monoclonal antibody and glyceraldehyde-3-phosphate dehydrogenase (GAPDH) rabbit polyclonal antibodies (Abclonal, Woburn, MA, USA) or polyclonal anti-TRAF3 antiserum (Beyotime, Shanghai, China). The following day, membranes were incubated with the secondary antibody. The visualization of antigen–antibody complexes using Western Bright ECL (Beyotime, Beijing, China) and images were analyzed using ImageJ (v. 1.53c).

### 2.9. Statistical Analysis

Statistical analysis was performed using GraphPad Prism (v. 8.0.2). All experiments were technically repeated at least three times, and the results are presented as the means ± standard error (SE). Relative gene expression levels were determined using the 2^−ΔΔCT^ method. Student’s *t*-test was employed to assess significant differences between groups, with a *p*-value < 0.05 considered statistically significant.

## 3. Results

### 3.1. Poly(I:C) Stimulation Induces Downregulation of miR-1388 Expression

Poly(I:C) treatment induces an antiviral response (Figure 1A,B). To investigate the role of miRNA in poly(I:C) stimulation, we analyzed the relative expression levels of miR-1388 in various tissues of silver carp after poly(I:C) stimulation. We found a significant downregulation of miR-1388 in all tissues except for the gills (Figure 1C). To further validate these findings, we examined the expression levels of miR-1388 in macrophages stimulated with different concentrations or fixed concentrations of poly(I:C) for various durations. We observed a concentration-dependent downregulation of miR-1388 (Figure 1D) as well as a time-dependent downregulation (Figure 1E) of miR-1388. These results suggest the involvement of miR-1388 in the innate immune process.

### 3.2. miR-1388 Has an Effect on the Antiviral Genes Stimulated by poly(I:C)

To investigate the role of miR-1388 in the innate immune response triggered by poly(I:C), we initially confirmed that transfection with miR-1388 mimics significantly increased the expression level of miR-1388 in macrophages (Figure 2A). Subsequently, we observed that only transfection with miR-1388 resulted in a slight but non-significant decrease in the expression levels of antiviral genes (Figure 2B), while a significant downregulation occurred after poly(I:C) stimulation (Figure 2C). To further validate these findings, we used the miR-1388 inhibitor and found that it significantly reduced the expression level of miR-1388 in macrophages (Figure 2D). Transfection with the miR-1388 inhibitor led to an elevation in the expression level of antiviral genes (Figure 2E), which exhibited a significant upregulation only after poly(I:C) stimulation (Figure 2F). These results demonstrate that miR-1388 plays a crucial role in the innate immune response triggered by poly(I:C).

### 3.3. miR-1388 Showed a Targeting Relationship with TRAF3

To elucidate the role of miR-1388 in innate immune response mechanisms, we predicted the potential target gene *TRAF3* for miR-1388 using TargetScan (v5.0) and miRanda (v3.3a) and found the targeting sequence of miR-1388 in the 3′UTR of *TRAF3* (Figure 3A). We co-transfected wild-type or mutant dual luciferase reporter plasmids with or without miR-1388 mimics and found that miR-1388 only inhibited the activity of the wild-type luciferase, with no effect on the mutant (Figure 3B). This inhibition was concentration-dependent (Figure 3C) and could be reversed by the addition of the miR-1388 inhibitor (Figure 3D). Additionally, we observed that miR-1388 significantly suppressed GFP expression in wild-type mVenus-*TRAF3*-3′UTR but not in the mutant. (Figure 3E). These results suggest a targeting relationship between miR-1388 and *TRAF3*.

### 3.4. TRAF3’s Role in Innate Immune Response and IRF3-Mediated Signaling Pathway Activation

Following poly(I:C) stimulation, *TRAF3* was significantly upregulated in various tissues of silver carp, with the kidney showing the most substantial increase (Figure 4A). The mRNA expression level of *TRAF3* in macrophages gradually increased with increasing concentrations of poly(I:C) (Figure 4B). Similarly, the protein level of TRAF3 showed a concentration-dependent increase (Figure 4C). When the concentration of poly(I:C) was fixed, it was found that the mRNA expression level of *TRAF3* initially increased with time, peaking at 6 h before gradually decreasing (Figure 4D). Compared to the empty vector (EV), the addition of the *TRAF3* expression plasmid significantly enhanced the activity of both *IRF3* and *ISRE* reporter genes, as indicated by luciferase activity (Figure 4E). These results suggest that *TRAF3* is involved in the innate immune response and activates the *IRF3*-mediated signaling pathway.

### 3.5. Correlation between miR-1388 Expression and TRAF3 Levels

To understand the regulatory role of miR-1388 on *TRAF3*, we co-transfected the *TRAF3* expression plasmid with or without miR-1388. It was found that miR-1388 significantly inhibited the mRNA expression of exogenous *TRAF3* (Figure 5A), and the same result was observed for the protein expression of exogenous TRAF3 (Figure 5B). Regarding endogenous *TRAF3*, we found that miR-1388 significantly suppressed the mRNA expression of endogenous *TRAF3* (Figure 5C) and exhibited a concentration-dependent inhibition of TRAF3 protein expression (Figure 5D). Conversely, the miR-1388 inhibitor markedly increased the mRNA expression of *TRAF3* (Figure 5E) and showed a concentration-dependent increase in TRAF3 protein expression (Figure 5F). These results indicate a negative relationship between miR-1388 and *TRAF3* expression.

### 3.6. miR-1388 Regulates Antiviral Genes Primarily through Its Target Gene TRAF3

miR-1388 significantly inhibited the luciferase activity of the *IRF3* and *ISRE* reporter genes compared to the control group (Figure 6A), which were upregulated following poly(I:C) stimulation (Figure 6B). As the concentration of si-TRAF3 increases, the protein level of endogenous TRAF3 gradually decreases. (Figure 6C). Silencing TRAF3 resulted in a significant reduction in the expression of antiviral genes compared to the control group (Figure 6D). These results indicate that miR-1388 primarily regulates antiviral genes by targeting *TRAF3*.

## 4. Discussion

Viral diseases pose significant challenges in fish farming, with innate immunity serving as the primary defense mechanism against invading pathogens in teleost fish. Among the various components of innate immunity, *IFN-I* plays a pivotal role. In recent years, several miRNAs implicated in fish innate immunity, such as miR-133b and miR-731 [23,24], have been extensively characterized and functionally analyzed. Nevertheless, our comprehension of the regulatory mechanisms governing miRNAs in fish remains limited, impeding further advancements in this field of study. In previous investigations, our research team discovered the involvement of miR-1388 in regulating the innate immunity of silver carp [25]. In the present study, we aimed to elucidate the specific role played by miR-1388 in poly(I:C) stimulation, thereby furnishing a solid theoretical foundation for understanding the intricate network of miRNA-mediated host–pathogen interactions. Ultimately, we anticipate that our findings will contribute to improved guidance for aquaculture practices.

Numerous studies have demonstrated the crucial role of *TRAF3*, a multifunctional immune molecule, in antiviral innate immunity [26,27]. Acting as the adaptor protein that links MAVS to mitochondria, TRAF3 facilitates K63-linked ubiquitination, thereby activating IFN-I [11]. The level of TRAF3 ubiquitination also holds significance in the antiviral process [28]. Mice lacking TRAF3 show lower immune responses and higher mortality rates [29]. The deletion of TRAF3 in macrophages and dendritic cells significantly diminishes *IFN-I* production [30,31]. Our study revealed a significant upregulation of *TRAF3* in various tissues following poly(I:C) stimulation, with the most substantial increase observed in the kidney, followed by the spleen—both being major immune organs. Similar effects were also observed in macrophages, underscoring the pivotal role of TRAF3 in immunity. TRAF3 has been found to participate in the signaling pathways mediated by *TLR3*, *MDA5*, and *RIG-I*, thereby promoting *IFN-I* secretion [32]. Additionally, our study documented the significant activation of key molecules, such as the *IRF3* promoter and *ISRE*, in the *IFN* signaling pathway by TRAF3 in common carp. Furthermore, following the small interfering RNA-mediated silencing of TRAF3, a notable decrease in the expression of antiviral genes and *IFN-I* was observed in macrophages. This observation indicates the involvement of TRAF3 in the *IRF3*-mediated signaling pathway, highlighting its significance as a pivotal, though not exclusive, molecule necessary for *IFN-I* production. In this study, we demonstrated that *TRAF3* is a target of miR-1388, providing insights into the mechanism of the miRNA-mediated regulation of antiviral responses in teleost fish.

miRNAs have been demonstrated to have a close association with viral infections [33]. Research indicates that miRNAs can directly target viral genes, thereby regulating viral replication and controlling viral infections [34,35]. Consequently, virus-induced alterations in host miRNA expression or the production of beneficial miRNAs can enable the evasion of host immune recognition and clearance, thereby facilitating immune evasion [36,37]. Furthermore, miRNAs have the capability to target host immune genes, activate immune-related pathways, and consequently modulate the host immune response [38,39]. Poly(I:C), an artificial synthetic viral mimic, mimics viral infection and induces the expression of miR-489 in miiuy croaker, thereby mediating innate immunity [40]. Previous studies have highlighted the significant role of miR-1388 in antiviral responses in bony fish [41]; however, relevant research on the antiviral mechanism of miR-1388 remains scarce. In our investigation, miR-1388 was identified as one of the poly(I:C)-responsive miRNAs and was found to regulate the expression of *ISGs* and *IFN-I* in silver carp macrophages. To the best of our knowledge, this is the first study to explore the role of poly(I:C)-stimulated miRNA in silver carp and the initial documentation of the crucial role of miR-1388 in regulating *ISGs* and *IFN-I* in silver carp macrophages. Our study illustrates that miR-1388 modulates the antiviral response in silver carp by targeting *TRAF3*. The negative feedback mechanisms within organisms are essential for maintaining internal equilibrium. Excessive antiviral immune responses triggered by viral invasions can lead to a “cytokine storm”, resulting in host damage and potential organ failure. Specific miRNAs, such as miR-126, miR-217, and miR-218 [42,43,44], play a role in negatively modulating cytokine levels to uphold normal immune balance. In miiuy croaker, miR-1388-5p has been validated to negatively regulate the NF-κB pathway by targeting *IRAK1* [45]. It is noteworthy that, owing to the effective regulation of miRNAs on immune homeostasis, some studies suggest that miRNAs may enhance vaccine efficacy and serve as biomarkers for adverse reactions and prognosis post-vaccination [46,47]. MiRNAs have been utilized in the development of attenuated live vaccines for influenza viruses, enterovirus 71, and other pathogens [48,49,50]. Nonetheless, research on fish miRNA vaccines is limited, and due to their lack of side effects, they hold potential significance in the prevention of aquatic diseases.

## 5. Conclusions

In summary, our research has identified miR-1388 as a regulator of the natural immunity in silver carp. Following poly(I:C) stimulation, the expression of miR-1388 is downregulated, leading to the promotion of IFN-I and antiviral gene expression by targeting *TRAF3*. On the contrary, miR-1388 in exogenous cells can also inhibit the activation of TRAF3 on *IRF3* and *ISRE* reporter genes. In conclusion, we demonstrate that *TRAF3* is a novel target of miR-1388, which enhances innate immunity by regulating TRAF3 expression to maintain immune system stability. Our study contributes significantly to the understanding of miR-1388 and provides a novel theoretical basis for investigating the intricate miRNA regulatory networks in aquatic organisms.

## Figures and Tables

**Figure 1 biomolecules-14-00694-f001:**
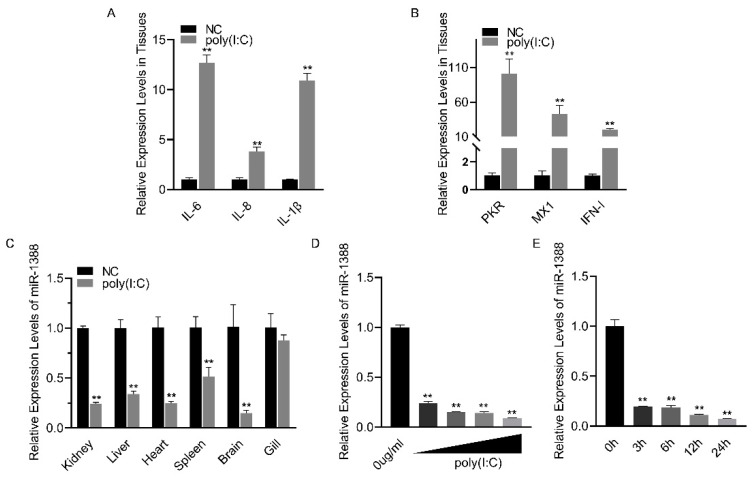
Poly(I:C) stimulation induces the downregulation of miR-1388 expression. (**A**,**B**) The mRNA expression levels of antiviral genes in the kidney tissues of silver carp treated with poly(I:C) were measured using qRT-PCR. (**C**) The expression level of miR-1388 in various tissues of silver carp stimulated by poly(I:C) was determined by qRT-PCR. (**D**) The expression level of miR-1388 in macrophages treated with different concentrations of poly(I:C) was assessed by qRT-PCR. (**E**) The expression level of miR-1388 in macrophages following treatment with poly(I:C) for varying durations was examined using qRT-PCR. Each experiment was replicated at least three times, and the results are presented as the means ± SE. **, *p* < 0.01 versus the control group.

**Figure 2 biomolecules-14-00694-f002:**
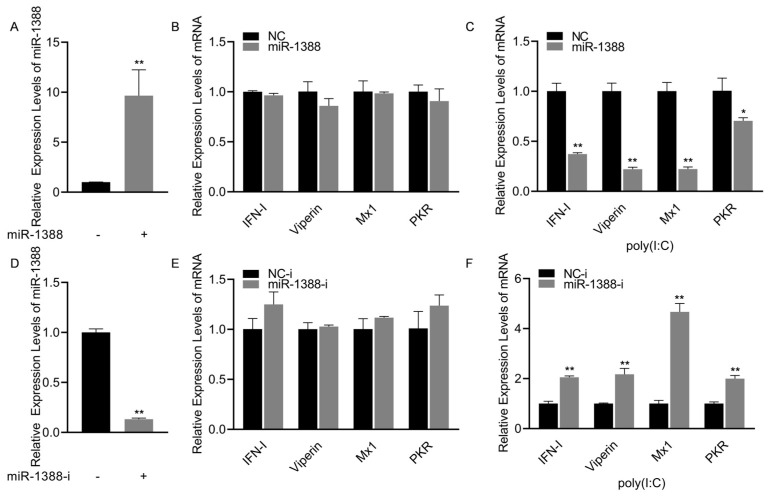
miR-1388’s impact on the antiviral genes stimulated by poly(I:C). (**A**) Following a 48 h transfection of miR-1388 mimics, the expression level of miR-1388 in macrophages was quantified using qRT-PCR. (**B**) The expression levels of antiviral genes in macrophages were assessed by qRT-PCR after a 48 h transfection with mimics. (**C**) Additionally, macrophages were transfected with mimics for 48 h and then stimulated with poly(I:C) for 6 h to examine the impact on antiviral gene expression, which was measured using qRT-PCR. (**D**) The expression level of miR-1388 in macrophages was determined by qRT-PCR following the 48 h transfection of the miR-1388 inhibitor. (**E**) The effect of the inhibitor on antiviral gene expression was evaluated by measuring their levels in macrophages using qRT-PCR after a 48 h transfection. (**F**) Moreover, macrophages transfected with the inhibitor for 48 h were stimulated with poly(I:C) for 6 h to assess its influence on antiviral gene expression, which was quantified by qRT-PCR. All experiments were performed at least three times, and the results are presented as the means ± SE. Statistical analysis revealed significant differences compared to the control group: *, *p* < 0.05; **, *p* < 0.01.

**Figure 3 biomolecules-14-00694-f003:**
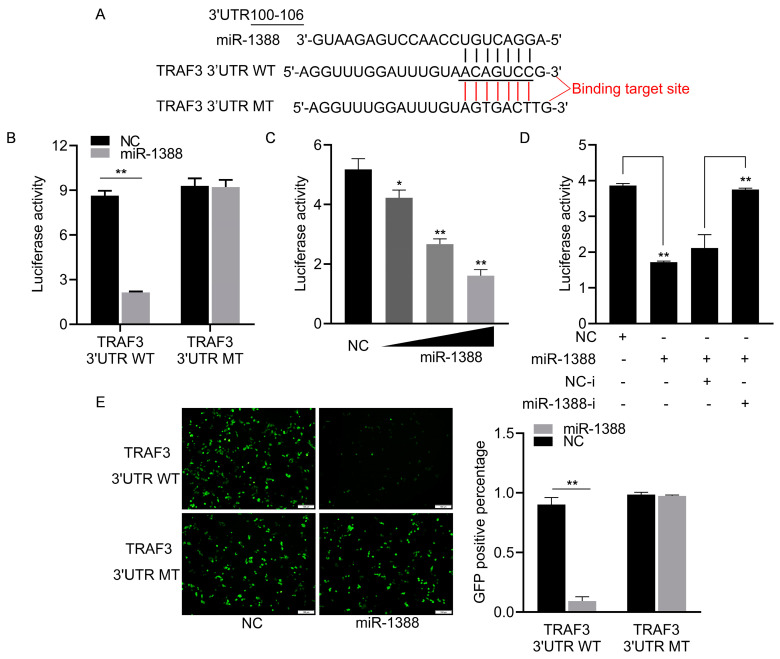
Analysis of miR-1388 targeting *TRAF3* 3′UTR. (**A**) Utilization of bioinformatics tools to predict potential binding sites of miR-1388 in the 3′UTR of *TRAF3*. (**B**) Co-transfection of wild-type or mutant *TRAF3* 3′UTR dual-luciferase reporter plasmids with miR-1388 or NC into HEK293T cells. Subsequently, luciferase activity was assessed after 48 h. (**C**) Co-transfection of wild-type *TRAF3* 3′UTR dual-luciferase reporter plasmids with varying concentrations of miR-1388 (0, 25, 50, 100 nM) or NC (100, 50, 25, 0 nM) into HEK293T cells. After 48 h, luciferase activity was measured. (**D**) Co-transfection of wild-type *TRAF3* 3′UTR dual-luciferase reporter plasmids with miR-1388, miR-1388-i, NC, NC-i into HEK293T cells. Following 48 h incubation, luciferase activity was quantified. (**E**) Co-transfection of wild-type or mutant mVenus-*TRAF3*-3′UTR constructs with miR-1388 or NC into HEK293T cells. Post 48 h, fluorescence intensity was evaluated using an enzyme-linked instrument. The scale bar is set at 100 μm, and the original magnification is ×10. All luciferase activities were normalized to the Renilla luciferase activity. Each experiment was conducted at least thrice, and the results are presented as the means ± SE. *, *p* < 0.05; **, *p* < 0.01 compared to the control group.

**Figure 4 biomolecules-14-00694-f004:**
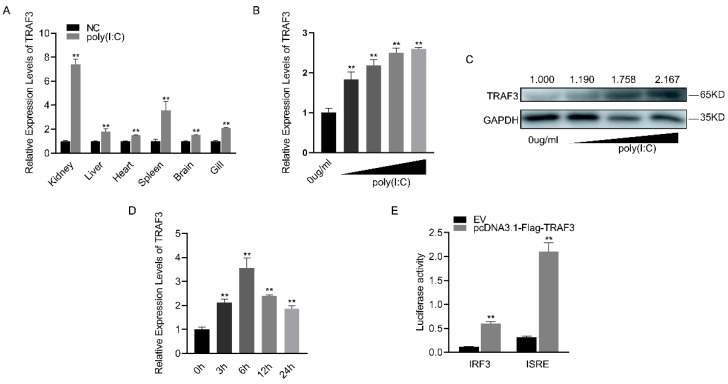
Expression levels and functions of *TRAF3* in innate immune responses. (**A**) The mRNA expression levels of *TRAF3* in various tissues of silver carp were detected using RT-qPCR after 6 h of poly(I:C) stimulation. (**B**) The mRNA expression levels of *TRAF3* in macrophages were detected using RT-qPCR after stimulation with different concentrations of poly(I:C). (**C**) The protein expression levels of TRAF3 in macrophages were detected using Western blotting after stimulation with different concentrations of poly(I:C). (**D**) The mRNA expression levels of *TRAF3* in macrophages were detected using RT-qPCR at different time points (0, 6, 12, 24 h) after poly(I:C) stimulation. (**E**) Luciferase activity was measured after the co-transfection of *IRF3*, *ISRE* luciferase reporter plasmid with *TRAF3* overexpression plasmid or EV in HEK293T cells for 48 h. All luciferase activities were normalized to the Renilla luciferase activity. Each experiment was conducted at least three times, and the results are presented as the means ± SE. **, *p* < 0.01 compared to the control group. Original images can be found in Appendix A.

**Figure 5 biomolecules-14-00694-f005:**
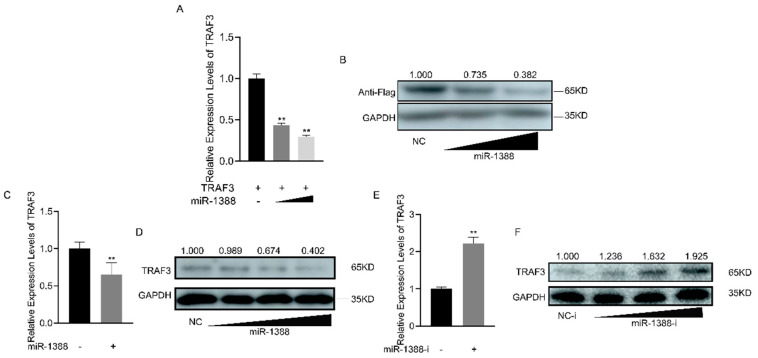
Expression analysis of the miR-1388 and *TRAF3* interaction. (**A**) The mRNA expression levels of *TRAF3* in HEK293T cells were determined using RT-qPCR after co-transfection with different concentrations of miR-1388 or NC for 48 h. (**B**) The protein expression levels of TRAF3 in HEK293T cells were assessed following co-transfection with different concentrations of miR-1388 or NC for 48 h. (**C**) The mRNA expression levels of *TRAF3*- in macrophages were measured using RT-qPCR after transfection with miR-1388 or NC for 48 h. (**D**) The protein expression levels of TRAF3 in macrophages were evaluated after transfection with different concentrations of miR-1388 or NC for 48 h. (**E**) The mRNA expression levels of *TRAF3*- in macrophages were quantified using RT-qPCR after transfection with miR-1388-i or NC-i for 48 h. (**F**) The protein expression levels of TRAF3 in macrophages were analyzed following transfection with different concentrations of miR-1388-i or NC-i for 48 h. Each experiment was performed at least three times, and the results are presented as the means ± SE. **, *p* < 0.01 compared to the control group. Original images can be found in Appendix A.

**Figure 6 biomolecules-14-00694-f006:**
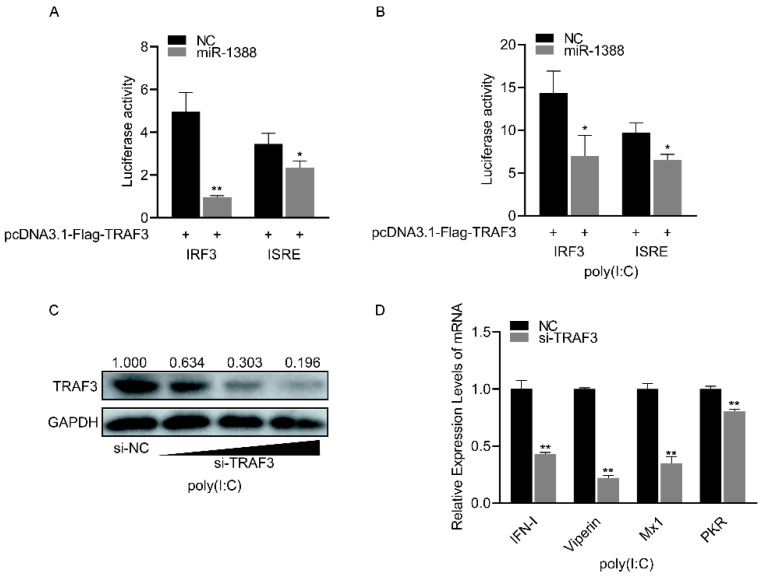
Regulation of antiviral genes by miR-1388 via *TRAF3* targeting. (**A**) Luciferase activity of *IRF3* and *ISRE* luciferase reporter plasmids co-transfected with *TRAF3* overexpression plasmid, miR-1388, or NC was measured in HEK293T cells after 48 h. (**B**) Luciferase activity of *IRF3* and *ISRE* luciferase reporter plasmids co-transfected with *TRAF3* overexpression plasmid, miR-1388, or NC was measured in HEK293T cells after 48 h, followed by stimulation with poly(I:C) for 6 h. (**C**) Assessment of TRAF3 protein expression levels in macrophages after transfection with an increasing amount of si-*TRAF3* or NC for 48 h. (**D**) mRNA expression levels of antiviral genes in macrophages were determined using RT-qPCR after transfection with si-TRAF3or NC for 48 h, followed by stimulation with poly(I:C) for 6 h. All luciferase activities were normalized to Renilla luciferase activity. Each experiment was performed at least three times, and results are presented as means ± SE. *, *p* < 0.05; **, *p* < 0.01 compared to the control group. Original images can be found in Appendix A.

## Data Availability

Data are contained within the article and Appendix A.

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
