# Peer review of "Downregulation of miR-1388 Regulates the Expression of Antiviral Genes via Tumor Necrosis Factor Receptor (TNFR)-Associated Factor 3 Targeting Following poly(I:C) Stimulation in Silver Carp (Hypophthalmichthys molitrix)"

_biomolecules, 2024, doi:10.3390/biom14060694_

Round 1

Reviewer 1 Report

Comments and Suggestions for Authors

This is a well-written article in which the authors investigate the role of miR-1388 in the antiviral signaling pathway activated by TRAF3 post poly(I:C) stimulation, to enhancing understanding of the mechanisms through which fish miRNAs participate in immune responses.

.

The manuscript is interesting and worth publication, however, I have some observations:

1.     Material and metods: In the RNA extraction and real-time quantitative PCR Section insert the method used to measure gene expression and include in Table the amplification efficiency, Tm, and product size for all primers sets.

2.     Figure 1 B: “The mRNA expression levels of antiviral genes in the kidney tissues of silver carp treated with poly(I:C) were measured using qRT-PCR”. give an explanation for such high values.

3.     Figure 2A e 2D It is not clear why is indicated protein expression.

Reviewer 2 Report

Comments and Suggestions for Authors

In this manuscript, Gao et al. report that miR-1388 is down-regulated following poly(I:C) stimulation, indicating this miRNA plays a role in suppressing immune activation. At the molecular biology level, the authors elucidate that miR-1388 targets to 3’UTR of TRAF3 gene and down-regulates its transcription. Both overexpression of miR-1388 and suppression of TRAF3 reduce the transcription of anti-viral genes. The experiments were well designed. Here are 2 detail points:

-There are small text mistakes in the manuscript. Line 306, one comma needs to be removed. Line 334, aimed needs to be changed to black.

- To confirm that miR-1338 downregulates anti-viral genes via targeting to TRAF3, a rescue experiment is needed. For instance, if knockdown TRAF3 on top of the miR-1338 inhibitor, can depletion of TRAF3 rescue the increase of antiviral genes caused by miR-1388 inhibitor?

Reviewer 3 Report

Comments and Suggestions for Authors

Dear Editor,

The manuscript entitled “Downregulation of miR-1388 regulates the expression of antiviral genes via TRAF3 targeting following poly(I:C) stimulation in silver carp (Hypophthalmichthys molitrix)” by Kun Gao et al. presents an evaluation of the regulatory influence of miR-1388 on the IRF3-mediated signaling pathway by targeting TRAF3 post poly(I:C) stimulation. The authors report that following stimulation with poly(I:C) miR-1388 expression was significantly upregulated in primary immune tissues and macrophages. A dual luciferase reporter assay indicated a direct targeting relationship between miR-1388 and tumor necrosis factor receptor (TNFR)-associated factor 3 (TRAF3) while a distinct negative post-transcriptional correlation between miR-1388 and TRAF3 was demonstrated, including significant negative post-transcriptional regulatory association between miR-1388 and the levels of antiviral genes following poly(I:C) stimulation. Reporter plasmids were also used to elucidate the role of miR-1388 in the antiviral signaling pathway and siRNA-TRAF3 confirmed that miR-1388 regulates the expression of antiviral genes and the production of type I interferons (IFN-I) through its interaction with TRAF3.

Τhe manuscripts’ objectives are quite interesting, defining specific biological problems in a very comprehensive way and applying specific assays to answer them. The manuscript is well-written and could be accepted for publication after minor revisions. My detailed comments for the authors to consider are provided below:

1.      Page 2, line 54: please define abbreviations.

2.   Page 2, line 88: please specify which tissues were collected instead of “various”.

3.  Page 3, lines 97-98: Please add a brief description of gradient centrifugation or add an appropriate reference.

4.      Page 6, figure 2A: Shouldn’t the Y axis title be expression levels instead of protein levels?

5.    Page 8, figure 4C: This image seems to be a western blot while the legend describes an RT-qPCR assay. Please clarify what data you intended to use and correct the figure and legend accordingly.

6.      Page 10, figure 6C: To my understanding, this image does not contain what is described in the legend and in the text (i.e. si-TRAF3-1, si-TRAF3-2, si-TRAF3-3) but an increasing amount of si-TRAF3? Please clarify what is described and correct the figure and legend accordingly.

Round 2

Reviewer 3 Report

Comments and Suggestions for Authors

The provided information are adequate for me and the manuscript could be accepted for publication.